# A Longitudinal Study in Turkiye of Host Ability to Produce Antibodies following a Third Homologous BNT162b2 Vaccination

**DOI:** 10.3390/vaccines11040716

**Published:** 2023-03-23

**Authors:** Mustafa Genco Erdem, Ozge Unlu, Mehmet Demirci

**Affiliations:** 1Department of Internal Medicine, Faculty of Medicine, Beykent University, Istanbul 34398, Turkey; 2Department of Medical Microbiology, Faculty of Medicine, Istanbul Atlas University, Istanbul 34403, Turkey; 3Department of Medical Microbiology, Faculty of Medicine, Kirklareli University, Kirklareli 39100, Turkey

**Keywords:** surrogate neutralizing antibody, anti-S-RBD IgG titers, obesity, BNT162b2

## Abstract

Obesity is a multifaceted, complex condition that has negative impacts on one’s health. There are conflicting reports regarding the COVID-19 vaccine’s ability to induce antibody formation in obese people. Our study aimed to determine anti-S-RBD IgG and surrogate neutralizing antibody (snAb) levels before and after the third Pfizer–BioNTech (BNT162b2) vaccination (at 15, 60, 90, and 120 days) in normal-weight adults, overweight, and obese individuals without any comorbidity or previous SARS-CoV-2 infection history, but it did not evaluate the response to the first two doses. In this longitudinal prospective study in Istanbul, Turkey, a total of 323 consecutive adult individuals (141 normal weight, 108 overweight, and 74 patients with obesity) were included. Peripheral blood samples were collected. Anti-S-RBD IgG and surrogate neutralizing antibody levels were detected using the ELISA method. After the third dose of BNT162b2 vaccination, obese patients had significantly lower levels of snAb against SARS-CoV-2 compared with normal-weight controls, but the levels otherwise did not differ between the study groups. Across all individuals in our cohort, titers peaked about a month after this third vaccination and then gradually faded. Anti-S-RBD IgG and snAb IH% levels against SARS-CoV-2 were not correlated with IL-6 and TNF-α levels. In conclusion, anti-S-RBD IgG titers and snAb IH% levels against SARS-CoV-2 were determined longitudinally for 120 days after the third homologous BNT162b2 vaccination. Although there were no significant differences in anti-S-RBD IgG, we found significant differences in the snAb IH% levels against SARS-CoV-2 between obese and healthy control subjects.

## 1. Introduction

Nearly 2 billion individuals worldwide are affected by the obesity pandemic, which is defined as having a body mass index of 30 or more [1]. Obesity is a complicated, multifaceted condition that is known to have adverse effects on health due to excessive body fat. A significant increase in the rate of obesity has been observed worldwide in the last 50 years [2]. Obesity increases the risk of many serious diseases, such as type 2 diabetes mellitus (T2DM), cardiovascular diseases (CVD), and nonalcoholic fatty liver disease (NAFLD). It places a significant financial load on the health systems of various nations [2,3]. The COVID-19 pandemic caused by SARS-CoV-2 is still ongoing. Obese or overweight individuals are more susceptible to the serious complications that develop with COVID-19 [4]. In obesity, the adaptation feature of the human body to adapt to overweight and its direct effects, such as excessive inflammation, and indirect effects, such as insulin resistance or hyperglycemia, are considered to be important risk factors for the course of COVID-19 disease [5]. Adiposity plays an important role in the host’s response to infection due to its role in altering the innate and acquired immune systems.

Physiologically, adipose tissue is essential for normal metabolism and the immune system; however, “meta-inflammation”, which results from excessive adiposity and exceeding the normal fatty acid storage threshold, disrupts mitochondrial functions and apoptotic signals and increases the production of reactive oxygen species (ROS) [1,6]. Hypertrophic adipocytes secrete proinflammatory cytokines such as TNF-α and IL-6. These two cytokines contribute to the decreased insulin sensitivity observed in obesity. TNF-α increases the concentration of circulating free fatty acids (FFAs) and directly blocks the transmission of insulin signals. IL-6 increases the production of acute-phase reactants such as C-reactive protein (CRP) and contributes to increased insulin resistance. TNF-α and IL-6 increase lipolysis and decrease triglyceride synthesis. As a result, both cytokines may release FFAs into the bloodstream and increase their accumulation in the liver, skeletal muscle, and pancreatic β cells [7]. Obesity contributes to decreased efficacy of different vaccines such as hepatitis B, influenza, and rabies [8]. However, there are conflicting studies regarding the effectiveness of the COVID-19 vaccines in obese individuals, stating that either their efficacy has diminished or not [9,10,11]. In our study, we aimed to determine the anti-S-RBD IgG antibody titers and surrogate neutralizing antibody (snAb) levels before and after the third homologous Pfizer–BioNTech (BNT162b2) vaccine (on days 15, 60, 90, and 120) in normal-weight, overweight, and obese individuals without any comorbidity or history of SARS-CoV-2 infection but did not evaluate the response to the first two doses.

## 2. Materials and Methods

### 2.1. Study Design and Samples Collection

In this longitudinal prospective study, we included a total of 323 consecutive individuals (age >18 years) (141 normal weight, 108 overweight, and 74 patients with obesity) who were admitted to the internal medicine clinic and vaccination clinic in the private MedicalPark hospital in Istanbul, Turkey. We selected these adults randomly after a two-dose BNT162b2 vaccination and before a third BNT162b2 vaccination dose between March and July, 2022. These participants did not have diabetes mellitus and were asked to complete an additional questionnaire regarding comorbidity and previous SARS-CoV-2 infections. They were also comprehensively assessed by the physician based on age, gender, and comorbidity. Exclusion criteria were: (I) age <18 years or >50 years old, (II) a history of comorbidities, (III) an active infection, and (IV) pregnant women. The third vaccination dose was received in a median of 154 (IQR [Interquartile Range 25–75 percentiles]: 102–165) days after the second vaccination dose in this study. Informed written consent was obtained from all participants prior to the study. Ethical approval for this study was obtained from the Ethics Committee of the Kirklareli University Medicine Faculty (Approval number: E3784467719940823). Electronic health records, such as routine laboratory data, were extracted from hospitals’ routine health information systems. The Panbio COVID-19 (Abbott, Chicago, IL, USA) rapid antigen test kit was used according to the manufacturer’s instruction to investigate an active SARS-CoV-2 infection in the participants at each blood draw. Peripheral blood samples were collected directly before the third vaccination (0th day). Follow-up blood samples were drawn on the 15th, 60th, 90th, and 120th day after the third vaccination dose. All peripheral blood samples were transported to the laboratory and were processed on the same day as collection.

### 2.2. IL-6 and TNF-α Testing

The IL-6 and TNF-α concentrations from the serum samples were evaluated using the enzyme-linked immunosorbent assay (ELISA) according to the manufacturer’s instructions (Elabscience, Wuhan, China). Absorbance was taken at 450 nm wavelength using ELx800 microplate reader (BioTek Instruments, Berlin, Germany). The assay detection ranges IL-6 and TNF-α were 1.56~100 pg/mL and 7.81~500 pg/mL, respectively.

### 2.3. Antibody Testing

Serum samples were obtained from peripheral blood samples using centrifugation at 400× *g* for 10 min at room temperature in the laboratory. The Abbott Architect SARS-CoV-2 immunoglobulin G (IgG) test (Abbott, Chicago, IL, USA) detects IgG antibodies against the nucleocapsid protein (NCP) of SARS-CoV-2 in accordance with the manufacturer’s instructions in order to detect past SARS-CoV-2 infection in all participants. The participants with a concentration above the median (IQR) of 2.03 signal-to-cutoff (S/Co) ratios were considered to have a prior infection with SARS-CoV-2. To quantitatively detect IgG antibodies against the receptor-binding region (RBD) of the spike protein S1 subunit of SARS-CoV-2, the Abbott Architect SARS-CoV-2 IgG II Quant test (Abbott, Chicago, IL, USA) was used according to the manufacturer’s instruction. Results were evaluated as arbitrary units per milliliter (AU/mL). To convert to the “Binding Antibody Unit per milliliter (BAU/mL)” in the WHO’s International Standard, results were multiplied by the correlation coefficient of 0.142 [12].

To detect the surrogate neutralizing antibody (snAb) against SARS-CoV-2, the SARS-CoV-2 NeutraLISA assay (Euroimmun, Lübeck, Germany) was used as the snAb assay using the competitive ELISA method according to the manufacturer’s instruction. Results were evaluated as snAb percent inhibition (IH%). Tests with an snAb IH% ≥ 35% were evaluated as positive, and tests with an snAb IH% < 20 were considered negative [13,14].

### 2.4. Statistical Analysis Methods

The IBM Corp. Released 2011. IBM SPSS Statistics for Windows, Version 20.0. Armonk, NY: IBM Corp. was used for statistical analyses. Continuous data was described with mean and standard deviation; categorical data was expressed with frequencies (N) and percentages (%). Comparisons between groups were analyzed using the Kruskal–Wallis test. After statistically significant results, post hoc comparisons were performed using the Bonferroni-corrected Mann–Whitney U test. Repeated measures ANOVA tests and the Bonferroni-corrected Tukey test were performed and comparisons made within subjects, group × time, and between subjects. The correlation was evaluated with Spearman’s correlation coefficient (r_s_): r_s_ < 0.25 was evaluated as not statistically correlated, r_s_ = 0.25–0.5 was evaluated as a weak correlation, r_s_ = 0.5–0.75 was evaluated as a moderate correlation, r_s_ = 0.76–0.85 was evaluated as a strong correlation, and r_s_ > 0.85 was evaluated as a very strong correlation. The statistical significance threshold was considered to be *p* < 0.05.

## 3. Results

When the age and gender distributions of the normal-weight controls and the overweight and obese individuals included in the study were examined, no statistically significant difference was found between the groups (*p* > 0.05). However, a statistical difference was found in the laboratory data obtained from their blood before the third dose of BNT162b2 vaccination (0th day) (*p* < 0.05). Insulin and HOMA-IR levels were highest in the obese patient group (Table 1).

The IL-6 levels in the obese patient group were 2 and 10 times higher than in the overweight and normal-weight control groups, respectively. The TNF-α levels were 1.5 and 1.7 times higher in obese patients compared with overweight and normal-weight control groups, respectively.

Before the third dose of BNT162b2 vaccination (0th day), anti-S-RBD IgG II antibody levels were 4946.20 ± 3368.23, 4801.80 ± 3470.82, and 4417.20 ± 2756.08 AU/mL in the normal-weight control, overweight, and obese groups, respectively (*p* > 0.05) (Table 2). Although the third dose of BNT162b2 increased the anti-S-RBD IgG II antibody approximately 5- and 3.5-fold in all three study groups at day 15 and day 60 after vaccination, respectively, there was no statistical difference between the groups (*p* > 0.05). At 90 days after the third dose of BNT162b2 vaccination, the anti-S-RBD IgG II antibody level was found to be significant in obese patients compared with the normal-weight control and overweight groups (*p* < 0.05). Table 2 shows the repeated measures ANOVA tests. After the repeated measures ANOVA tests, no difference was found within subjects, group × time, and between subjects for anti-S-RBD IgG levels. However, repeated measures ANOVA tests showed a significant difference between subjects for snAb (IH%). The Bonferroni-corrected Tukey test was performed and found healthy control vs. overweight *p* = 0.774, healthy control vs. obese *p* = 0.008, and overweight vs. obese *p* = 0.038. Only healthy control vs. obese (*p* = 0.008) was found to be significantly different.

Although comparison between subjects was not significant (*p* ^3^ = 0.209) (Table 2), there was a clear trend for the obese group to have lower antibody response than the two nonobese groups, as can be seen in Figure 1b.

As a result of the post hoc evaluation, a statistically significant difference was found only between the obese and overweight groups in terms of the anti-S-RBD IgG variable at day 90. (Bonferroni correction was applied; *p* < 0.0166 was accepted as the significance limit.) In terms of the snAb variable, a statistically significant difference was found only between the obese and healthy control groups at day 15. (Bonferroni correction was applied; *p* < 0.0166 was accepted as the significance limit.)

At day 90 after the third dose of BNT162b2 vaccination, the anti-S-RBD IgG antibody titers decreased by approximately 55% in the normal-weight control and overweight groups compared with day 15, while a 65% decrease was found in the obese patient group. At day 120 after the third dose of BNT162b2 vaccination, there was no statistical difference between the groups for anti-S-RBD IgG antibody titers (*p* > 0.05).

When the percentage of inhibition of the surrogate neutralizing antibody against SARS-CoV-2 (snAb IH%) was examined in the study groups, no statistical difference was found between the groups before the third dose of BNT162b2 vaccination (0th day) (*p* > 0.05). However, on the 15th day of the third dose of BNT162b2 vaccination, snAb IH% in obese patients was significantly different from the normal-weight control and overweight groups (*p* < 0.05) (Table 2). On day 15, the snAb IH% level was lower in the obese group compared with the normal-weight controls and the overweight study group. When the snAb IH% levels were followed up on days 60, 90, and 120 after the third dose of BNT162b2 vaccination, no statistical difference was found between the study groups (*p* > 0.05) (Table 2). Figure 1a,b and Figure 2a,b show the anti-S-RBD IgG and snAb IH% levels detected in the study groups against SARS-CoV-2.

Anti-S-RBD IgG antibody titers and snAb IH% levels against SARS-CoV-2 were not correlated with IL-6 and TNFα levels (*p* > 0.05).

## 4. Discussion

Obesity is a multifaceted, complex condition that has negative impacts on one’s health [2]. Furthermore, there are conflicting reports in studies on the antibody efficacy of COVID-19 vaccines in obese individuals, but it is still thought that there is an increased risk of COVID-19 with obesity [9,10,11,15]. Therefore, the goal of our study was to assess the anti-S-RBD antibody and surrogate neutralizing antibody (snAb) levels in obese people before and after the third BNT162b2 immunization (at 15, 60, 90, and 120 days) and compare them to those in normal-weight controls and overweight people.

Piernas et al. reported that vaccine efficacy was 86% after the second vaccination in obese individuals (40 kg/m^2^ and more) in phase 3 trials and was higher than in low-weight individuals, but stated that the risk of severe COVID-19 was still higher in this group. They emphasized that this may be due to impaired T-cell responses but reported this limitation in their data because they could not follow up for a long time [9]. Butsch et al. reported that The Obesity Society concluded that COVID-19 vaccines on the market had good efficacy without weight differences and that their efficacy was not significantly different in people with and without obesity [10]. Kara et al. reported that obese patients who had not had COVID-19 and had a history of COVID-19 infection had significantly lower anti-S-RBD antibody levels after CoronaVac and BNT162b2 vaccines compared with healthy and normal-weight individuals. The fact that both neutralizing antibody levels and HOMA-IR values were not measured, baseline antibody levels were not measured before vaccination, and no follow-up was performed are considered to be limitations of this study [11]. In our study, participants were followed up five different times for 120 days after the third dose of BNT162b2 vaccination, and anti-S-RBD IgG antibody titers against SARS-CoV-2 were measured. Although Zhang et al. reported that anti-S-RBD IgG antibody titers were associated with BMI, they reported that this difference may be due to inactivated vaccination and that there was no such difference in mRNA vaccines [15]. Similar to the results of our study, Bates et al. reported that anti-S-RBD IgG antibody titers were similar in obese, overweight, and normal-weight controls [16]. Yamamoto et al. reported that anti-S-RBD IgG antibody titers decreased with BMI in men but were similar in women [17].

Bates et al. [16] examined snAb levels in obese, overweight, and normal-weight controls on days 50 and 200 and reported that BMI had no significant impact on anti-S-RBD-IgG titers and snAb on day 50 [16]. Soffer et al. reported that BMI was associated with neutralizing antibody levels in COVID-19 patients and that neutralizing antibody levels were high in obese COVID-19 patients [18]. Hu et al. reported that neutralizing antibody levels against SARS-CoV-2 were low after inactivated SARS-CoV-2 vaccination [19]. Levin et al. reported that snAb levels were 31% higher in obese individuals compared with nonobese individuals at 6 months after the second dose of BNT162b2 vaccination. However, they stated that it was unclear whether they were at higher or lower risk from COVID-19 and whether the antibodies developed with the vaccine were protective [20].

Adipose tissues in obese patients contain M1 macrophages, which can frequently cause local and systemic inflammation and act as a source of proinflammatory cytokine production. Other defense cells such as neutrophils and dendritic cells similarly increase inflammation by contributing to the production of various proinflammatory cytokines. Therefore, as a consequence of obesity, a chronic inflammatory state at both the local and systemic levels is detected in patients. Inflammation is also at the forefront in SARS-CoV-2 infections, and the main complications of COVID-19 have been found to be directly related to systemic inflammation. There are studies showing that disease severity and dysregulation of proinflammatory cytokines interact directly in COVID-19 patients. It is possible that the inflammation caused by the combination of COVID-19 and obesity may increase the systemic inflammation present in obesity. All these inflammation inferences suggest that the immune system is more likely to overreact [21]. However, our data on the antibodies developed after vaccination did not support this.

Obesity is associated with an increased risk of many common noncommunicable diseases such as diabetes mellitus (DM), cardiovascular diseases (CVD), metabolic diseases, cancers, and nonalcoholic fatty liver disease (NAFLD). This provides evidence that obese individuals often carry multiple comorbidities and may develop a severe susceptibility to viral infections such as COVID-19. It is also known that the presence of excess adipose tissue in obese patients can lead to an increase in angiotensin-converting enzyme 2 (ACE2) receptors and increased susceptibility to different viral and bacterial infections. Studies on influenza A virus in obese patients have shown that, in addition to increased disease severity, obesity increases the shedding time of the virus, thereby accelerating the transmission of the virus from person to person and contributing to the production of more infectious aerosols. This may be similar in SARS-CoV-2 infection [22]. It is thought that poor vaccine responses may develop in COVID-19 patients due to the effect of obesity on the immune system. The effect of obesity-induced inflammation on the immune system response is still not clearly understood. While regulatory and suppressive cytokines such as IL-4, IL-10, IL-13, and IL-33 are predominantly detected in the adipose tissue in healthy individuals, proinflammatory cytokines such as IL-1β, IL-6, IL-12, IL-18, TNF-α, and IFN-γ are detected in the adipose tissue of obese individuals. Both adipokines and cytokines are known to direct the development of the immune response against pathogens. The presence or absence of these immune system elements in individuals during vaccination may alter the development of the adaptive immune response, leading to differences in the levels of protective antibodies that develop following vaccination. Levels of the hormone leptin, an adipokine that regulates appetite and controls energy metabolism in obese patients, are known to be associated with responses to RNA virus vaccines such as influenza, so it is also thought that leptin may play a role in COVID-19 progression in the chronic inflammation seen in obesity patients. High leptin levels are also associated with decreased levels of Treg cells, which may lead to a shift in cytokine gene expression levels in a proinflammatory direction. Leptin may also contribute to inflammation by increasing T-cell proliferation by activating the JAK/STAT pathway. However, it is also suggested that resistance may develop at high leptin levels and the host may adapt to this [23]. Similarly, proinflammatory cytokines were found to be increased in our study.

Hyperinsulinemia, which can also be detected in obese patients, may cause SARS-CoV-2 uptake in adipocytes by upregulating the expression of GRP78, which acts as a binding cofactor of the viral spike (S) and cellular angiotensin-converting enzyme 2 (ACE2) receptor in the host. In addition, due to the low vaccine efficacy found after influenza A and hepatitis B virus vaccines, there are concerns that a low immune response may develop against SARS-CoV-2 in obesity patients and that the vaccine response may be low. In most of the studies, only the presence of anti-SARS-CoV-2 antibodies was investigated, but neutralizing antibody activities were not controlled [24]. Recently, similar to the results of our study, Faizo et al. found no statistically significant difference in vaccine efficacy in obese patients. However, they reported that the neutralizing capacity of antibodies produced by obese individuals was significantly reduced compared with the control group [24]. Longitudinal analysis in our study revealed that this was not significantly different.

In their study, Bénédicte et al. reported that the humoral response to COVID-19 vaccine was lower in patients with obesity and diabetes one month after the second dose compared with controls. These findings emphasized the need for post-vaccination serological controls, especially in high-risk populations [25]. In our study, such a decrease was not detected, and no significant difference was found.

The limitations of this study are its single-center design, performing COVID-19 infection histories according to rapid antigen tests, nucleocapsid antibody measurement, and participant declarations. However, control of obesity-related parameters such as HOMA-IR and insulin, equivalent gender and age matching within groups, and long-term follow-up are its advantages. In our study, another limitation is that different cytokine responses other than TNF-α and IL-6 and the host response at the cellular level in PBMC were not controlled due to budgetary reasons. We did not study any health outcomes, and the differences in blood measurement of antibody response surrogates may not translate to meaningful differences in infection, health, or mortality outcomes. Our study was insufficiently powered to detect differences in all the measured immune responses at every timepoint. However, the aggregate longitudinal data clearly shows a small but consistent depression of immune responses in the obese group compared with the nonobese groups. This depression was present at baseline, suggesting that the difference between these groups likely arose starting with the responses to the first or second vaccination, which we did not measure in this study.

## 5. Conclusions

In conclusion, there was no difference between the study groups compared with overweight and normal-weight controls in terms of the levels of anti-S-RBD IgG and snAb. Although there were no significant differences in anti-S-RBD IgG, we found a significant difference in snAb IH% levels against SARS-CoV-2 between the obese and healthy control subjects. We thought that anti-S-RBD IgG antibody titers showed a rapid decrease and snAb levels showed a slow increase in obese patients. The low efficacy of neutralizing antibodies developed after vaccination in obese patients is notable, and this may explain the sensitivity to COVID-19. Comprehensive, controlled, and lengthy follow-up studies are believed to be necessary because there are still inconsistent results concerning the effectiveness of the SARS-CoV-2 vaccine in obese people.

## Figures and Tables

**Figure 1 vaccines-11-00716-f001:**
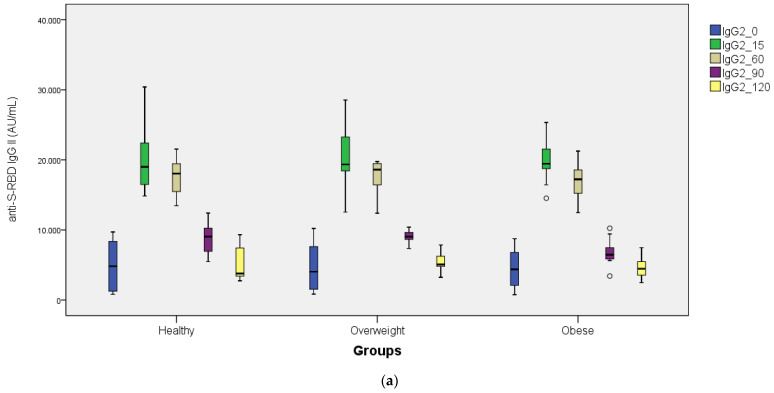
(**a**) Distribution of anti-S-RBD IgG antibody titers (AU/mL) before and after the third dose of BNT162b2 vaccine (on the 15th, 60th, 90th, and 120th days) for all groups (box plots). (**b**) Distribution of anti-S-RBD IgG antibody titers (AU/mL) before and after the third dose of BNT162b2 vaccine (on the 15th, 60th, 90th, and 120th days) for all groups (line plots).

**Figure 2 vaccines-11-00716-f002:**
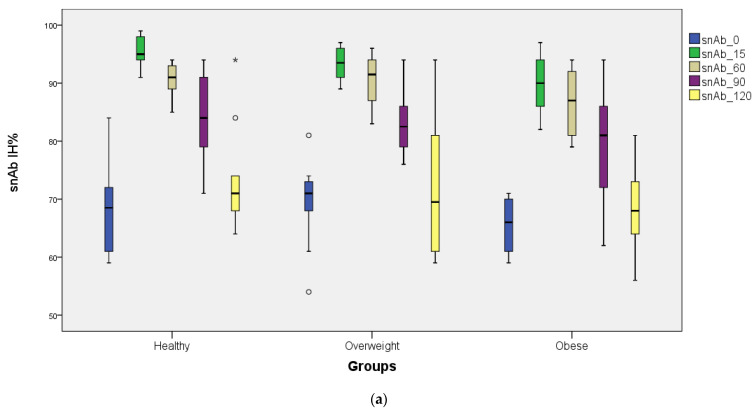
(**a**) Distribution of snAb IH% levels before and after the third dose of BNT162b2 vaccine (on the 15th, 60th, 90th, and 120th days) for all groups (box plots). (**b**) Distribution of snAb IH% levels before and after the third dose of BNT162b2 vaccine (on the 15th, 60th, 90th, and 120th days) for all groups (line plots).

**Table 1 vaccines-11-00716-t001:** Distribution of demographic and laboratory data in the individuals included in the study (Mean ± SD).

	Normal Controls (*n*: 141)	Overweight (*n*: 108)	Obese (*n*: 74)	*p* *
Age (years)	40.42 ± 8.17	42.64 ± 7.11	41.18 ± 8.14	0.295
Gender (M/F)	63/78	48/60	33/41	0.473
BMI (kg/m^2^)	22.62 ± 2.28	26.52 ± 1.11	30.85 ± 0.73	0.007
FBS (mg/dL)	85.7 ± 10.12	87.6 ± 7.88	92.47 ± 8.28	0.008
PPBS (mg/dL)	106.1 ± 11.31	124.22 ± 15.14	128.62 ± 24.27	0.006
HbA1c (%)	5.07 ± 0.34	5.39 ± 0.46	5.57 ± 0.42	0.008
Insulin (µUI/L)	6.12 ± 2.80	13.85 ± 7.62	15.48 ± 8.21	0.004
HOMA-IR	1.13 ± 0.88	2.93 ± 1.72	4.21 ± 2.13	0.007
ALT (U/L)	24.51 ± 10.87	31.43 ± 13.88	39.21 ± 18.72	0.003
AST (U/L)	18.54 ± 6.21	31.39 ± 15.12	35.51 ± 14.89	0.002
GGT (mg/dL)	31.22 ± 12.18	34.91 ± 13.63	36.19 ± 12.35	0.012
ALP (mg/dL)	98.37 ± 36.84	99.48 ± 41.68	102.11 ± 48.57	0.025
TG (mg/dL)	115.72 ± 26.16	124.43 ± 36.91	118.69 ± 38.33	0.015
HDL (mg/dL)	55.81 ± 13.78	46.89 ± 14.25	49.14 ± 13.18	0.009
LDL (mg/dL)	95.17 ± 16.93	121.16 ± 37.52	129.37 ± 35.64	0.007
IL-6 (pg/mL)	19.00 ± 15.32	89.10 ± 46.28	173.70 ± 93.11	0.003
TNF-α (pg/mL)	329.70 ± 89.92	361.40 ± 89.50	561.50 ± 154.66	0.002

* Kruskal–Wallis test. FBS: Fasting blood sugar. PPBS: Postprandial blood sugar.

**Table 2 vaccines-11-00716-t002:** Distribution of anti-S-RBD IgG and snAb IH% levels in all participants before (day 0) and after (days 15, 60, 90, and 120) the third vaccine dose (Mean ± SD).

	anti-S-RBD IgG II (AU/mL)					*p* ^1^	*p* ^2^	*p* ^3^
	0th	15th	60th	90th	120th			
Healthy controls (*n*: 141)	4946.20 ± 3368.23	20,505.20 ± 5380.14	17,736.30 ± 2771.17	9055.30 ± 2244.51	5209.90 ± 2434.32	0.001	0.982	0.209
Overweight (*n*: 108)	4801.80 ± 3470.82	20,348.00 ± 4329.38	17,647.80 ± 2370.54	9065.90 ± 897.56	5279.70 ± 1313.01			
Obese (*n*: 74)	4417.20 ± 2756.08	19,967.10 ± 3163.73	17,184.00 ± 2534.74	6778.30 ± 1936.71	4612.20 ± 1601.25			
	**snAb (IH%)**					***p*** **^1^**	***p*** **^2^**	***p*** **^3^**
	**0th**	**15th**	**60th**	**90th**	**120th**			
Healthy controls (*n*: 141)	69.70 ± 8.74	95.30 ± 2.71	90.70 ± 2.79	83.80 ± 7.54	73.10 ± 9.19	<0.001	0.997	0.007 *
Overweight (*n*: 108)	69.40 ± 7.38	93.30 ± 2.95	90.60 ± 4.03	83.40 ± 5.70	71.20 ± 11.79			
Obese (*n*: 74)	65.60 ± 4.95	89.70 ± 4.99	86.90 ± 5.86	79.00 ± 9.39	68.80 ± 7.67			

^1^ Within subjects, ^2^ group × time, ^3^ Between subjects, Bonferroni-corrected Tukey test, * Healthy control vs. Overweight *p* = 0.774, Healthy control vs. Obese *p* = 0.008, Overweight vs. Obese *p* = 0.038.

## Data Availability

No new data were created.

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
