# Peer review of "A Longitudinal Study in Turkiye of Host Ability to Produce Antibodies following a Third Homologous BNT162b2 Vaccination"

_vaccines, 2023, doi:10.3390/vaccines11040716_

Round 1

Reviewer 1 Report

Comments to the author:
The Manuscript "
Does obesity influence the host's ability to produce antibodies following the third homologous BNT162b2 vaccination? A longitudinal study. This study Indicates that in obese patients, snAb IH% levels slowly increased while Anti-S-RBD IgG antibody titters  rapidly decreased.

Major revisions required
Comments:
1- The obesity data was provided actually how much body weight and BMI in terms of whether subjects have metabolic syndrome or not

2- Study needs to be more refine in terms of statistics between subjects

3- this study shows that insulin secretion was so high but HbA1c is normal which is actually questionable?

Author Response

Thank you very much for your valuable comments and contribution. all comments have been carefully checked and edited on the manuscript.

You can also see our response below.

Thank you, Regards

Assoc. Prof. Mehmet Demirci

Response to Reviewer 1:

Comments to the author:
The Manuscript " Does obesity influence the host's ability to produce antibodies following the third homologous BNT162b2 vaccination? A longitudinal study”. This study Indicates that in obese patients, snAb IH% levels slowly increased while Anti-S-RBD IgG antibody titters  rapidly decreased.

Major revisions required
Comments:
1- The obesity data was provided actually how much body weight and BMI in terms of whether subjects have metabolic syndrome or not

Response: Thank you for your valuable comments and contributions. As stated in the manuscript, comorbidity history and data of all patients were also analysed. metabolic syndrome was not present in these patients.

2- Study needs to be more refine in terms of statistics between subjects

Response: Thank you for your valuable comments and contributions. on your suggestion, repeated measures ANOVA tests and Bonferroni Corrected Tukey test was performed within subjects, group x time, and between subjects comparisons.

3- this study shows that insulin secretion was so high but HbA1c is normal which is actually questionable?

Response: Thank you for your valuable comments and contributions. But This is the data of our patients.

Reviewer 2 Report

This study examines the effect of obesity on response to a third BNT162b2 COVID vaccination dose. The study is performed in a Turkish population. The study adds interesting literature review, data, analyses, and conclusions to the literature. It would be nice to see it eventually published. Currently, the data is misinterpreted and poorly presented. The discussion fails to emphasize the main conclusions. These issues need to be rectified before publication.

MAJOR ISSUES

The phenomenon here is that the obese response is marginally less than the non-obese response. This study is slightly underpowered to detect this lower response. So when it looks at four timepoints it sees nominal significance is some timepoints but not others. The overall conclusion is not that the response fluctuates but that it is consistently slightly less than the non-obese response. Therefore the suggestion that it fluctuates needs to be removed form the Abstract. And this entire issue needs to be prominent in the Discussion.

I think all of these issues would become clear if Figure 1 was replaced by a figure that looked like the attached figure (perhaps with error bars attached, but not necessary if still accompanied by Table 2). And Figure 2 replaced with a similarly designed figure. The attached figure makes it clear that the obese group is consistently lower than the other groups, but not by much.

Power calculations would help this discussion but are not strictly necessary if the authors do not have the resources/expertise to complete them.

The baseline of this study was after the second dose and before the third dose. Therefore this study did not seek to evaluate whether there were differences in vaccine response between obese and non-obese at the earlier doses. However, the Abstract and text are written as if there were NO differences from this earlier doses. The fact that the study did not evaluate response to the first two doses needs to be made clear in the Abstract and text body. That said, it does appear that at baseline, the response in obese is lower (Table2). I wonder if the data were fitted to a longitudinal regression (or some kind of graphical curve fit, this consistent trend across all timepoints from baseline to day 120 would become more clear.

TITLE

"Does obesity influence the host's ability to produce antibodies 2

following the third homologous BNT162b2 vaccination? A lon- 3 gitudinal study"

shorten it to make it a single phrase. Add the location to title. For example

"A longitudinal study in Turkey of host ability to produce antibodies following a third homologous BNT162b2 vaccination"

ABSTRACT

mention the study population somewhere (e.g., "in Turkey")

"Anti-S-RBD IgG and surrogate neutralizing antibody levels were de- 18 tected by the ELISA method."

if you are short on word count, you don't really need this in the Abstract. It can go in MEthods.

"at day 90 and day 15"

makes more sense to put these in order

"at day 15 and day 90"

but you need to mention what happened at day 60 and 120 in parallel, so something like"

at day 15 and day 90, but not at day 60 or 120"

and then, if this is true you need to explain why it fluctuated. Experimental error? Complex immune response dynamics? If the explanation does not fit in the abstract, then maybe the conflicting details don't either.

INTRODUCTION

"Many research- 40 ers accept that obese or overweight individuals are more susceptible to serious complica- 41 tions that develop with COVID-19"

just write for simplicity

"Obese or overweight individuals are more susceptible to serious complications that develop with COVID-19"

"Physiologically, adipose tissue is essential for normal metabolism and the immune 48 system; however, it is known that "Metainflammation," which results from excessive adi- 49 posity and exceeding the normal fatty acid storage threshold, disrupts mitochondrial 50 functions and apoptotic signals and increases the production of reactive oxygen species 51 (ROS) "

simpler as

"Physiologically, adipose tissue is essential for normal metabolism and the immune 48 system; however, "Metainflammation," which results from excessive adi- 49 posity and exceeding the normal fatty acid storage threshold, disrupts mitochondrial 50 functions and apoptotic signals and increases the production of reactive oxygen species 51 (ROS) "

"normal weight adults, overweight and obese individuals "

parallel structure

"normal weight, overweight and obese individuals "

"The third vaccination dose was en- 78 rolled in "

'enrolled' is the wrong word. Maybe 'given' or 'received', so

"The third vaccination dose was received at" 

"After 121 statistically significant results, Post Hoc comparisons were performed by using Bonferroni 122 corrected Mann-Whitney U test."

I don't understand this. Do you just mean

"Post Hoc comparisons were performed by using Bonferroni corrected Mann-Whitney U test."

If you mean something else, please write as two separate sentences. 

Table 1

please use no more significant figures than justified

so not

"40.42+8.17"

but rather

40.4 + 8

and I think you mean +/- not +

Define FBS an PPBS and all the other acronyms in the Table legend. 

"TNF-" appears to be missing a number or greek letter

Mention in the legend in Table 1 that all of these differences between groups were expected, based on stratification by weight into the three groups.

RESULTS

"4946.20+3368.23, 4801.80+3470.82, and 4417.20+2756.08 "

please be careful with significant figures

should be:

"4900+3400, 4800+3500, and 4400+2800"

"Although the third 144 dose of BNT162b2 increased approximately 5 and 3.5-fold in all three study groups at day 145 15 and day 60 after vaccination"

missing a word. perhaps you mean

"Although the third 144 dose of BNT162b2 increased anti-S-RBD IgG II antibody approximately 5 and 3.5-fold in all three study groups at day 145 15 and day 60 after vaccination"

TABLE 2

please pay attention to significant figures in all tables and in the main text of the paper. Not only to adhere to scientific standards but also to avoid clutter.

FIGURE 1 

ordinate labels appear to have European period rather than American comma. That is inconsistent with the text, which uses American conventions. Consider a convention like "40K" rather than "40.000". "40,000" is also OK but perhaps "40K" is less cluttered. Even better, just use "40" as the label next to the tick mark, and label the axis "anti-S-RBD IgG II (1000 AU/ml)".

DISCUSSION

Add to the limitations paragraph that you did not study any health outcomes, and that differences in blood measurement of antibody response surrogates may not translate to meaningful differences in infection, health, or mortality outcomes.

A number of recent articles (both peer reviewed and preprints) suggest a waning protection in all individuals after about 60 days (e.g., https://www.medrxiv.org/content/10.1101/2023.03.02.23286561v1). I think you cite some literature along these lines already. Although not strictly necessary, consider adding some of these more recent and/or more general publications about vaccine decline (e.g., those publications not specifically about obesity as a variable).

How do you view your results in light of these publications? Do you think obese people are driving the results in these other papers, and that non-obese have longer protection? Or do you think that perhaps protection is also waning in the non-obese people in your studied population and that you don't have the power to pick it up? 

Do you feel that your results are clinically actionable or actionable on a public health level to change current health practices?

SOMEWHERE

Make it clear you are studying the Pfizer vaccine. My preference would be to have that in the Abstract, but it could go elsewhere, depending on your style choices. e.g., write "Pfizer BNT162b2" in the abstract rather than "BNT162b2". This would help researchers using keywords to find your article.

HERE IS THE DATA FOR THE ATTACHED GRAPH

Time AU Arm

0 4.946 Healthy_Weight

15 20.505 Healthy_Weight

60 17.736 Healthy_Weight

90 9.055 Healthy_Weight

120 5.21 Healthy_Weight

0 4.802 Overweight

15 20.348 Overweight

60 17.648 Overweight

90 9.066 Overweight

120 5.28 Overweight

0 4.417 Obese

15 19.967 Obese

60 17.184 Obese

90 6.778 Obese

120 4.612 Obese

HERE ARE THE R COMMANDS FOR THE GRAPH

library(dplyr)

library(data.table)

library(ggplot2)

library(splines)

VO <- fread("/Users/xxx/Vaccines_obese_for_R_3_12_23.tsv", fill = TRUE)

VO$Arm <- factor(VO$Arm, levels = c("Healthy_Weight", "Overweight", "Obese"))

plot1 <- ggplot(data=VO, aes(x=Time, y=AU)) + geom_point(aes(colour=Arm)) + geom_line(aes(colour=Arm)) + labs(title="Vaccines Obesity Review", y = "anti-S-RBD IgG II (1000 AU/ml)", x = "Days After Third Dose")

plot1 + theme(legend.key.width=unit(1.1,"cm"))

Author Response

Thank you very much for your valuable comments and contribution. all comments have been carefully checked and edited on the manuscript.

You can also see our response below.

Thank you, Regards

Assoc. Prof. Mehmet Demirci

Response to Reviewer 2:

This study examines the effect of obesity on response to a third BNT162b2 COVID vaccination dose. The study is performed in a Turkish population. The study adds interesting literature review, data, analyses, and conclusions to the literature. It would be nice to see it eventually published. Currently, the data is misinterpreted and poorly presented. The discussion fails to emphasize the main conclusions. These issues need to be rectified before publication.

Response: Thank you for your valuable comments and contributions. 

MAJOR ISSUES

The phenomenon here is that the obese response is marginally less than the non-obese response. This study is slightly underpowered to detect this lower response. So when it looks at four timepoints it sees nominal significance is some timepoints but not others. The overall conclusion is not that the response fluctuates but that it is consistently slightly less than the non-obese response. Therefore the suggestion that it fluctuates needs to be removed form the Abstract. And this entire issue needs to be prominent in the Discussion.

Response: Thank you for your valuable comments and contributions. on your suggestion, The correction was carried out in the abstract.

I think all of these issues would become clear if Figure 1 was replaced by a figure that looked like the attached figure (perhaps with error bars attached, but not necessary if still accompanied by Table 2). And Figure 2 replaced with a similarly designed figure. The attached figure makes it clear that the obese group is consistently lower than the other groups, but not by much.

Response: Thank you for your valuable comments and contributions.  on your suggestion, Figure 1b and Figure 2b were created and added in the manuscript.

Power calculations would help this discussion but are not strictly necessary if the authors do not have the resources/expertise to complete them.

Response: Thank you for your valuable comments and contributions.  We could not perform any priori sample size calculation, because in the literature there wasn't any relevant reference according to obesity.

The baseline of this study was after the second dose and before the third dose. Therefore this study did not seek to evaluate whether there were differences in vaccine response between obese and non-obese at the earlier doses. However, the Abstract and text are written as if there were NO differences from this earlier doses. The fact that the study did not evaluate response to the first two doses needs to be made clear in the Abstract and text body. That said, it does appear that at baseline, the response in obese is lower (Table2). I wonder if the data were fitted to a longitudinal regression (or some kind of graphical curve fit, this consistent trend across all timepoints from baseline to day 120 would become more clear.

Response: Thank you for your valuable comments and contributions. On your suggestion, It was added to the abstract and text that the first two dose responses were not evaluated in our study.

TITLE

"Does obesity influence the host's ability to produce antibodies 2

following the third homologous BNT162b2 vaccination? A lon- 3 gitudinal study"

shorten it to make it a single phrase. Add the location to title. For example

"A longitudinal study in Turkey of host ability to produce antibodies following a third homologous BNT162b2 vaccination"

 Response: Thank you for your valuable comments and contributions. On your suggestion, Title was changed

ABSTRACT

mention the study population somewhere (e.g., "in Turkey")

 Response: Thank you for your valuable comments and contributions. On your suggestion, study population location was added.

"Anti-S-RBD IgG and surrogate neutralizing antibody levels were de- 18 tected by the ELISA method."

if you are short on word count, you don't really need this in the Abstract. It can go in MEthods.

 "at day 90 and day 15"

makes more sense to put these in order

"at day 15 and day 90"

but you need to mention what happened at day 60 and 120 in parallel, so something like"

at day 15 and day 90, but not at day 60 or 120"

and then, if this is true you need to explain why it fluctuated. Experimental error? Complex immune response dynamics? If the explanation does not fit in the abstract, then maybe the conflicting details don't either.

 Response: Thank you for your valuable comments and contributions. Repeated Measures ANOVA tests results between subjects was added in the abstract. 

INTRODUCTION

"Many research- 40 ers accept that obese or overweight individuals are more susceptible to serious complica- 41 tions that develop with COVID-19"

just write for simplicity

"Obese or overweight individuals are more susceptible to serious complications that develop with COVID-19"

  Response: Thank you for your valuable comments and contributions. On your suggestion, correction was made.

"Physiologically, adipose tissue is essential for normal metabolism and the immune 48 system; however, it is known that "Metainflammation," which results from excessive adi- 49 posity and exceeding the normal fatty acid storage threshold, disrupts mitochondrial 50 functions and apoptotic signals and increases the production of reactive oxygen species 51 (ROS) "

simpler as

"Physiologically, adipose tissue is essential for normal metabolism and the immune 48 system; however, "Metainflammation," which results from excessive adi- 49 posity and exceeding the normal fatty acid storage threshold, disrupts mitochondrial 50 functions and apoptotic signals and increases the production of reactive oxygen species 51 (ROS) "

   Response: Thank you for your valuable comments and contributions. On your suggestion, correction was made.

"normal weight adults, overweight and obese individuals "

parallel structure

"normal weight, overweight and obese individuals "

    Response: Thank you for your valuable comments and contributions. On your suggestion, correction was made.

"The third vaccination dose was en- 78 rolled in "

'enrolled' is the wrong word. Maybe 'given' or 'received', so

"The third vaccination dose was received at" 

Response: Thank you for your valuable comments and contributions. On your suggestion, correction was made.

"After 121 statistically significant results, Post Hoc comparisons were performed by using Bonferroni 122 corrected Mann-Whitney U test."

I don't understand this. Do you just mean

"Post Hoc comparisons were performed by using Bonferroni corrected Mann-Whitney U test."

If you mean something else, please write as two separate sentences. 

Response: Thank you for your valuable comments and contributions. On your suggestion, correction was made.

Table 1

please use no more significant figures than justified

so not

"40.42+8.17"

but rather

40.4 + 8

and I think you mean +/- not +

Response: Thank you for your valuable comments and contributions. On your suggestion, correction was made.

Define FBS an PPBS and all the other acronyms in the Table legend. 

Response: Thank you for your valuable comments and contributions.  On your suggestion, correction was made.

"TNF-" appears to be missing a number or greek letter

 Response: Thank you for your valuable comments and contributions.  On your suggestion, correction was made. 

Mention in the legend in Table 1 that all of these differences between groups were expected, based on stratification by weight into the three groups.

 RESULTS

"4946.20+3368.23, 4801.80+3470.82, and 4417.20+2756.08 "

please be careful with significant figures

should be:

"4900+3400, 4800+3500, and 4400+2800"

 Response: Thank you for your valuable comments and contributions.  On your suggestion, correction was made. 

"Although the third 144 dose of BNT162b2 increased approximately 5 and 3.5-fold in all three study groups at day 145 15 and day 60 after vaccination"

missing a word. perhaps you mean

"Although the third 144 dose of BNT162b2 increased anti-S-RBD IgG II antibody approximately 5 and 3.5-fold in all three study groups at day 145 15 and day 60 after vaccination"

 Response: Thank you for your valuable comments and contributions.  On your suggestion, correction was made. 

TABLE 2

please pay attention to significant figures in all tables and in the main text of the paper. Not only to adhere to scientific standards but also to avoid clutter.

FIGURE 1 

ordinate labels appear to have European period rather than American comma. That is inconsistent with the text, which uses American conventions. Consider a convention like "40K" rather than "40.000". "40,000" is also OK but perhaps "40K" is less cluttered. Even better, just use "40" as the label next to the tick mark, and label the axis "anti-S-RBD IgG II (1000 AU/ml)".

 Response: Thank you for your valuable comments and contributions.  Table 2 was changed after repeated measures ANOVA tests. 

DISCUSSION

Add to the limitations paragraph that you did not study any health outcomes, and that differences in blood measurement of antibody response surrogates may not translate to meaningful differences in infection, health, or mortality outcomes.

Response: Thank you for your valuable comments and contributions.  On your suggestion, correction was made. 

A number of recent articles (both peer reviewed and preprints) suggest a waning protection in all individuals after about 60 days (e.g., https://www.medrxiv.org/content/10.1101/2023.03.02.23286561v1). I think you cite some literature along these lines already. Although not strictly necessary, consider adding some of these more recent and/or more general publications about vaccine decline (e.g., those publications not specifically about obesity as a variable).

 Response: Thank you for your valuable comments and contributions.  

How do you view your results in light of these publications? Do you think obese people are driving the results in these other papers, and that non-obese have longer protection? Or do you think that perhaps protection is also waning in the non-obese people in your studied population and that you don't have the power to pick it up? 

 Response: Thank you for your valuable comments and contributions. As a result of repeated measures ANOVA analyses, the following conclusions were drawn from our study. The low efficacy of neutralising antibodies developed after vaccination in obese patients is notable and this may explain the sensitivity to COVID-19.

Do you feel that your results are clinically actionable or actionable on a public health level to change current health practices?

Response: Thank you for your valuable comments and contributions.  No, due to the limitations of being a single-centre study, we think that our data should be supported by comprehensive and multi-centre studies.

SOMEWHERE

Make it clear you are studying the Pfizer vaccine. My preference would be to have that in the Abstract, but it could go elsewhere, depending on your style choices. e.g., write "Pfizer BNT162b2" in the abstract rather than "BNT162b2". This would help researchers using keywords to find your article.

Response: Thank you for your valuable comments and contributions.  On your suggestion, correction was made.

HERE IS THE DATA FOR THE ATTACHED GRAPH

Time AU Arm

0 4.946 Healthy_Weight

15 20.505 Healthy_Weight

60 17.736 Healthy_Weight

90 9.055 Healthy_Weight

120 5.21 Healthy_Weight

0 4.802 Overweight

15 20.348 Overweight

60 17.648 Overweight

90 9.066 Overweight

120 5.28 Overweight

0 4.417 Obese

15 19.967 Obese

60 17.184 Obese

90 6.778 Obese

120 4.612 Obese

Response: Thank you for your valuable comments and contributions.  New figure was added with your suggestion.

HERE ARE THE R COMMANDS FOR THE GRAPH

library(dplyr)

library(data.table)

library(ggplot2)

library(splines)

VO <- fread("/Users/xxx/Vaccines_obese_for_R_3_12_23.tsv", fill = TRUE)

VO$Arm <- factor(VO$Arm, levels = c("Healthy_Weight", "Overweight", "Obese"))

plot1 <- ggplot(data=VO, aes(x=Time, y=AU)) + geom_point(aes(colour=Arm)) + geom_line(aes(colour=Arm)) + labs(title="Vaccines Obesity Review", y = "anti-S-RBD IgG II (1000 AU/ml)", x = "Days After Third Dose")

plot1 + theme(legend.key.width=unit(1.1,"cm"))

Response: Thank you for your valuable comments and contributions.  New figure was added with your suggestion.

Reviewer 3 Report

The manuscript shows the results obtained in a longitudinal study on the possible influence of obesity on the host's ability to produce antibodies after the third BNT162b2 homologous vaccination.

The work, although it presents interesting results, has several limitations that do not allow us to theorize about possible explanations of the results as to why obese individuals had a different anti-S-RBD IgG II and nAb response, especially a reduced elevation of anti -S. - RBD IgG II compared to the other groups.

The authors should have evaluated other cytokines in addition to IL-6 and TNF alpha, to get a better idea of the pro/anti-inflammatory profile. It would also have been very interesting to have evaluated the response at the cellular level in PBMC, including in-vitro stimulation studies with viral antigens. I suggest adding these limitations in the manuscript and proposing it for future studies.

I ask the authors if they could still incorporate into this study at least two or three additional cytokines, for example, IL-10, TGF beta and IL-4. With these new results they could round off a slightly more complete discussion.

Author Response

Thank you very much for your valuable comments and contribution. all comments have been carefully checked and edited on the manuscript.

You can also see our response below.

Thank you, Regards

Assoc. Prof. Mehmet Demirci

Response to Reviewer 3:

The manuscript shows the results obtained in a longitudinal study on the possible influence of obesity on the host's ability to produce antibodies after the third BNT162b2 homologous vaccination.

The work, although it presents interesting results, has several limitations that do not allow us to theorize about possible explanations of the results as to why obese individuals had a different anti-S-RBD IgG II and nAb response, especially a reduced elevation of anti -S. - RBD IgG II compared to the other groups.

The authors should have evaluated other cytokines in addition to IL-6 and TNF alpha, to get a better idea of the pro/anti-inflammatory profile. It would also have been very interesting to have evaluated the response at the cellular level in PBMC, including in-vitro stimulation studies with viral antigens. I suggest adding these limitations in the manuscript and proposing it for future studies.

Response: Thank you for your valuable comments and contributions. on your suggestion, this is indicated in the limitations section.

I ask the authors if they could still incorporate into this study at least two or three additional cytokines, for example, IL-10, TGF beta and IL-4. With these new results they could round off a slightly more complete discussion.

Response: Thank you for your valuable comments and contributions. on your suggestion, this is indicated in the limitations section. We would have liked to check many more cytokine responses, but we were unable to do so due to budgetary constraints.

Round 2

Reviewer 1 Report

it can be accepted in present form

Author Response

Thank you very much for your valuable comments and contribution. All comments have been carefully checked and edited on the manuscript.

You can also see our response below.

Thank you, Regards

Assoc. Prof. Mehmet Demirci

Response to Reviewer 1:

it can be accepted in present form.

Response: Thank you for your valuable comments and contributions. 

Reviewer 2 Report

The manuscript is much improved. There are still a few fundamental improvements which should be made. These are:

"On day 90 of the third homologous vaccination dose, the anti-S- 23 RBD IgG antibody titers decreased by approximately 55% in the normal weight control and over- 24 weight groups compared to day 15, while a 65% decrease was found in the obese patient group."

Remove this sentence from the abstract.

Normally I recommend precise inclusion of numbers in the abstract.

But because this study is underpowered, such precision at particular days is unwaranted. with your new final sentence, you adequately summarize your results. Let readers interested in the details read the whole manuscript, so they can get the full context for these figures.

You could replace this sentence with something like, "Across all individuals in our cohort, titers peaked about a month after this third vaccination, and then gradually faded."

Table 2

too many significant figures makes this table cluttered.

No reason to specify precision to the level of detail that you do.

In the legend of Table 2, note something along the lines of "although P3=0.209 is not significant, there is a clear trend for the obese group to have lower antibody response than the two non-obese groups, as can be seen in Figure 1b."

In the legend of table 2, clarify exactly what you mean by "within subjects" "group x time" and "between subjects". I believe the only interesting thing to test is whether the groups differ between each other. In which case the only thing of interest to the reader is P3 (if it means "between groups"). So don't even bother to report P1 and P2. 

Discussion

"It was found that there was a significant decrease in anti-S- 231 RBD IgG antibody titers in obese patients only on the 90th day after vaccination. "

Delete these sentences. The reason you don't see significance at particular timepoints is because your study was underpowered.

"Bates et al. reported that nAb levels in obese, overweight, and normal weight controls 239 were similar to our study [16]. The fact that follow-up was performed only on days 50 and 240 200 in this study may be the reason why anti-S-RBD IgG antibody titers on day 90 and 241 nAb level on day 15 were not found to be different compared to our study"

Delete these sentences. The reason you don't see significance at particular timepoints is because your study was underpowered.

Add to your last "limitations" paragraph:

Our study was insufficiently powered to detect differences in all measured immune responses at every timepoint. However, the aggregate longitudinal data clearly shows a small but consistent depression of immune responses in the obese group compared to the non-obese groups. This depression was present at baseline, suggesting that the difference between these groups likely arose starting with the responses to the first or second vaccination, which we did not measure in this study.

Conclusion

"it was found that obese patients included in our study after the third 322 dose of BNT162b2 vaccination had significantly lower anti-S-RBD IgG antibody titers at 323 day 90 and nAb levels at day 15."

Delete this sentence. It is not important. It draws attention to the spuriously fluctuating findings of significance between timepoints that is due to the underpowered study, and not to biology.

Author Response

Thank you very much for your valuable comments and contribution. All comments have been carefully checked and edited on the manuscript.

You can also see our response below.

Thank you, Regards

Assoc. Prof. Mehmet Demirci

Response to Reviewer 2:

The manuscript is much improved. There are still a few fundamental improvements which should be made. These are:

Response: Thank you for your valuable comments and contributions. 

"On day 90 of the third homologous vaccination dose, the anti-S- 23 RBD IgG antibody titers decreased by approximately 55% in the normal weight control and over- 24 weight groups compared to day 15, while a 65% decrease was found in the obese patient group."

Remove this sentence from the abstract.

Normally I recommend precise inclusion of numbers in the abstract.

But because this study is underpowered, such precision at particular days is unwaranted. with your new final sentence, you adequately summarize your results. Let readers interested in the details read the whole manuscript, so they can get the full context for these figures.

You could replace this sentence with something like, "Across all individuals in our cohort, titers peaked about a month after this third vaccination, and then gradually faded."

Response: Thank you for your valuable comments and contributions.  On your suggestion, The correction was carried out in the manuscript at abstract part and replaced the sentence.

 Table 2

too many significant figures makes this table cluttered.

No reason to specify precision to the level of detail that you do.

In the legend of Table 2, note something along the lines of "although P3=0.209 is not significant, there is a clear trend for the obese group to have lower antibody response than the two non-obese groups, as can be seen in Figure 1b."

Response: Thank you for your valuable comments and contributions. On your suggestion, The correction was carried out in the manuscript

In the legend of table 2, clarify exactly what you mean by "within subjects" "group x time" and "between subjects". I believe the only interesting thing to test is whether the groups differ between each other. In which case the only thing of interest to the reader is P3 (if it means "between groups"). So don't even bother to report P1 and P2.

Response: Thank you for your valuable comments and contributions. Yes, what we mean by between subjects is between groups. Since both the request of another referee and the information from the statistician was in favour of presenting this table in this way, it was not changed.

Discussion

"It was found that there was a significant decrease in anti-S- 231 RBD IgG antibody titers in obese patients only on the 90th day after vaccination. "

Delete these sentences. The reason you don't see significance at particular timepoints is because your study was underpowered.

Response: Thank you for your valuable comments and contributions. On your suggestion, The correction was carried out in the manuscript in the discussion part

"Bates et al. reported that nAb levels in obese, overweight, and normal weight controls 239 were similar to our study [16]. The fact that follow-up was performed only on days 50 and 240 200 in this study may be the reason why anti-S-RBD IgG antibody titers on day 90 and 241 nAb level on day 15 were not found to be different compared to our study"

Delete these sentences. The reason you don't see significance at particular timepoints is because your study was underpowered.

Response: Thank you for your valuable comments and contributions. On your suggestion, we deleted these sentences and correction was carried out in the manuscript in the discussion part

Add to your last "limitations" paragraph:

Our study was insufficiently powered to detect differences in all measured immune responses at every timepoint. However, the aggregate longitudinal data clearly shows a small but consistent depression of immune responses in the obese group compared to the non-obese groups. This depression was present at baseline, suggesting that the difference between these groups likely arose starting with the responses to the first or second vaccination, which we did not measure in this study.

Response: Thank you for your valuable comments and contributions. On your suggestion,  we added these paragraph on limitation part.

 Conclusion

"it was found that obese patients included in our study after the third 322 dose of BNT162b2 vaccination had significantly lower anti-S-RBD IgG antibody titers at 323 day 90 and nAb levels at day 15."

Delete this sentence. It is not important. It draws attention to the spuriously fluctuating findings of significance between timepoints that is due to the underpowered study, and not to biology..

Response: Thank you for your valuable comments and contributions. On your suggestion, we deleted these sentences

Reviewer 3 Report

The manuscript was improved and the answers to the questions were sufficient and accepted.

Author Response

Thank you very much for your valuable comments and contribution. All comments have been carefully checked and edited on the manuscript.

You can also see our response below.

Thank you, Regards

Assoc. Prof. Mehmet Demirci

Response to Reviewer 3:

The manuscript was improved and the answers to the questions were sufficient and accepted.

Response: Thank you for your valuable comments and contributions.